## MINI REVIEW

# Uncovering biology by single-cell proteomics

M. Shahid Mansuri[1], Kenneth Williams[1] & Angus C. Nairn [2✉]

Recent technological advances have opened the door to single-cell proteomics that can answer key biological questions regarding how protein expression, post-translational modifications, and protein interactions dictate cell state in health and disease.

I n-depth transcriptomic and proteomic analyses and understanding the mechanisms that control mRNA and protein expression at the tissue and cell level is key to characterizing normal and pathological biology. Next-generation sequencing technology (NGS) has been widely used to assess gene expression. To detect and quantify low levels of nucleic acid, RNA sequencing methods overcome the problem of low mRNA abundance by making multiple copies of the target molecules through polymerase chain reaction (PCR). Nucleic acid amplification coupled with barcoding has allowed the development of high-throughput genomic and transcriptomic methods to profile mRNA at the single cell or single nuclei level[1–3]. However, protein abundance cannot be scaled up due to the lack of methods such as PCR, resulting in a major obstacle to carrying out proteomics at the single-cell level. This is important since while single-cell transcriptome analyses provide information about cell phenotype, protein expression dictates cell state. Moreover, understanding how protein abundance, protein–protein interactions, and protein post-translational modifications are altered by diseases and responses to therapeutic drugs should lead to improved treatments[2,3]. Biochemists are therefore keenly interested in the single-cell analysis of proteins because analyzing tissues that are composed of diverse cell types hides cell-to-cell differences, making it difficult to interpret the resulting data. Hence, new methods are needed to enable proteome analysis at the single-cell level.

### Challenges and opportunities

There are several antibody and fluorescence-based methods that detect a few proteins at a time in individual cells, thus providing some degree of single-cell resolution[4–6]. Ideally, however, a technology is needed that can identify and quantify the abundance of all proteins in an individual cell. One study reported that the mammalian proteome is present at about $10^{10}$ total protein molecules per cell for cell lines with characteristic volumes of 2–4000 [$\mu m^3$][7]. Protein abundance in mammalian cells varies over a $>10^5$ range from ~4000 copies/cell for some transcription factors to ~$1.5 \times 10^8$ copies/cell for actin, with a median expression level of ~170,000 copies/cell[8]. Of the 19,116 proteins encoded in the human genome[9], about 10–12,000 are "housekeeping" proteins that are ubiquitously expressed in all cells[10]. Currently, proteomic technologies rely on "bottom-up" mass spectrometry-based methods to identify and quantitate proteolytically derived peptides. Given that the sensitivity of mass spectrometers has been reported to be as low as <75 zeptomole for individual peptides[11], researchers have realized that there are no technological barriers to single-cell proteomics (SCP). However, despite continued improvements in instrumentation, a major challenge is isolating single-cell contents, digesting the extracted proteomes to completion with protease, transferring the resulting peptides efficiently to the mass spectrometer without any sample loss, and analyzing the data obtained.

### Recent advances in sample handling and data analysis

To meet the challenge of sample depletion during preparation, there have been a number of recent advances in what is now a rapidly moving field of study of SCP, with important

[1] Yale/NIDA Neuroproteomics Center and Department of Molecular Biophysics and Biochemistry, Yale School of Medicine, New Haven, Connecticut, USA. [2] Yale/NIDA Neuroproteomics Center and Department of Psychiatry, Yale School of Medicine, New Haven, Connecticut, USA. ✉email: angus.nairn@yale.edu

contributions from several investigators. Earlier work combined capillary electrophoresis (CE) with high-resolution mass spectrometry (HRMS) to analyze proteomes of single blastomeres from *Xenopus laevis* embryos[12]. More recently, Zhu et al. developed a method called NanoPOTS (nanodroplet processing in one pot for trace samples)[13]. This method reduces liquid-handling steps and increases the yield of proteins per cell resulting in 2- to 25-fold more peptides being identified than when samples were prepared in small tubes. They demonstrated that NanoPOTS robustly identified ~1500–3000 proteins from 10 to 140 cells and further increased proteomic depth up to >3000 proteins by utilizing the Match Between Runs algorithm in the MaxQuant software used to analyze the data[13]. Similarly, in another study, the NanoPOTS approach enabled about 670 protein groups to be identified in extracts from single HeLa cells[14].

Nikolai Slavov's team took a different approach by developing a method called SCoPE-MS (single-cell proteomics by mass spectrometry) that sought to reduce sample losses caused by adhering to chromatographic columns or other surfaces by including a carrier proteome that increases the amount of low abundance peptides subjected to mass spectrometry. Notably, in this method, the test and carrier samples are labeled individually with isobaric tags and then pooled, which is a technically challenging step for small-volume samples[15]. However, the carrier sample, which has 200-fold higher total protein abundance than the single cells and is labeled with one isobaric tandem mass tag (TMT) reagent, can trigger MS2 and thus enable quantification of single cells in the TMT set based solely on their reporter ion signal[15]. SCoPE-MS analyses of 12 single Jurkat and 12 single U-937 cells identified a total of 767 proteins[15]. In the case of SCoPE-MS[15], while the inclusion of a carrier proteome enables more peptides to be identified, this may negatively influence quantitation[16]. This results from the difficulty in carrying out accurate MS quantitation on the very low signals from peptides derived from proteins in the single cells in the presence of the very large peptide signals from the carrier proteins[17]. To better understand this challenge, Cheung and coworkers analyzed the relationship between the level of carrier proteome and the quantitative accuracy of SCoPE-MS[16]. The results showed that an increase in carrier proteome amount required a corresponding increase in the number of ions sampled to maintain quantitative accuracy. Based on the results obtained, they provided guidance on SCP experimental design, data collection, and data analysis and developed a program, SCPCompanion, that enables quality-control analysis of SCP-MS data.

While Cheung et al. discuss the need for appropriate MS acquisition parameters when carrying out SCP studies, it remains unclear which levels of excess carrier provide the optimal balance between sensitivity and accuracy or the extent to which reduced ratios impair quantitative precision[18]. Therefore, Ctortecka et al. carried out a comprehensive study of currently available multiplexed SCP technologies that considered protein identifications, measurement variance, quantitative accuracy, and missing data. They conclude that limiting carrier spikes to ≤20-fold is critically important for accurate SCP analyses and that DIA-TMT analyses provide improved replicate overlap as compared to DDA-TMT analyses while preserving acquisition throughput at an improved quantitative accuracy. Please see further discussion of DIA and DDA below.

An optimized workflow, SCoPE2[19,20], incorporated several improvements over SCoPE-MS that were directed at increasing the accessibility, throughput, and accuracy[20]. Improvements included (1) lysing cells with the Minimal ProteOmic sample Preparation (mPOP) approach that uses a freeze-heat cycle that extracts proteins in water, thus obviating cleanup prior to liquid

chromatography mass spectrometry (LC-MS) analysis, (2) decreasing the lysis and digest volumes from 10 to 1 µl which increases the trypsin and substrate protein concentrations in the digest by 10-fold, (3) use of a reference channel comprised of aliquots from all sample sets, (4) a shorter nLC gradient that increases throughput and MS sensitivity, and (5) systematic parameter optimization that incorporated improvements previously developed to help optimize the acquisition of MS data (i.e., DO-MS; Data-driven Optimization of MS)[21] and to enhance peptide identifications in the acquired SCP LC-MS/MS data (i.e., DART-ID; Data-driven Alignment of Retention Times for Identification)[22]. As explained in Specht et al., accurate quantitation of LC-MS/MS data requires sampling the apex of chromatographic peaks[20]. Thus, Huffman et al. developed DO-MS to optimize the instrument parameters needed to enable chromatographic peaks to be sampled typically within 3 s of their apexes[21]. In addition, SCoPE2 improved peptide sequence identification by using a Bayesian algorithm (DART-ID) to incorporate retention time information that increases the confidence of assigning peptide sequences to MS/MS spectra while rigorously determining the false discovery rate (FDR)[22]. The high spectral purity and the use of retention times to bolster peptide sequence identification allowed assigning peptide sequences to about 35% of the MS2 spectra from each SCoPE2 run at 1% FDR[20].

Further studies to improve sample preparation include the work of Schoof et al., who developed an improved workflow for automated SCP. They found that including 20% trifluoroethanol (TFE) during lysis and 10% TFE during trypsin digestion increased the number of identified proteins by about 5%[6]. They also carried out studies to optimize MS instrument settings for SCP. These studies showed that injection times of 300 and 500 ms provided a good balance between proteome depth and quantitative performance[6]. Finally, SCeptre (Single Cell proteomics readout of expression), which is a python package that extends the capabilities of Scanpy to process single-cell MS/MS data, was described by Schoof et al.[6]. As detailed here, https://scanpy.readthedocs.io/en/stable/, Scanpy is a scalable, Python-based toolkit for analyzing single-cell gene expression data. SCeptre takes result files from Proteome Discoverer and meta information from the individual cells, which is a key feature that enables SCeptre to link recorded FACS parameters back to each cell. The authors found that the total summed intensity per cell provided a suitable parameter to detect outlier cells that could originate from duplets, empty wells, or sample loss during preparation. Filtering out such cells was crucial to avoid misleading biological conclusions[6]. Using their improved workflow, analysis of eight 384-well plates of single cells from a primary leukemia model system identified 2723 proteins across 2035 cells with an average of 987 proteins identified per cell.

A limitation of data-dependent label-free quantification (DDA) is the presence of missing values that complicates the analysis of even technical replicates. This challenge is exacerbated by the lower signal/noise ratios characteristic of SCP LC-MS/MS data. Kalxdorf et al. tried to address this challenge by developing the IceR (Ion current extraction Re-quantification) workflow that combines high identification rates of DDA with low missing value rates similar to data-independent acquisition (DIA)[23]. These authors applied IceR to SCP data and showed that it increased the number of proteins identified reliably in single-cell samples.

A miniaturized filter-aided sample preparation system termed MICROFASP was developed to carry out the parallel processing of 146 single blastomeres isolated from 50-cell stage *Xenopus laevis* embryos (~200 ng of protein/blastomere)[24]. Based on

LC-MS/MS analyses on 109 blastomeres, an average of 3468 ± 229 protein groups and 14,525 ± 2437 unique peptides were identified from each blastomere, with 1311 proteins being identified in every blastomere[24]. Brunner et al. also recently described a single-cell workflow that combines miniaturized sample preparation, very low flow rate (100 nl/min) chromatography, and a novel trapped ion mobility mass spectrometer (TIMS), which resulted in 10-fold improved sensitivity over 1 μl/min flow rate chromatography[25]. The method, termed true single-cell-derived proteomics (T-SCP), requires precise sample handling where without loss, single cells are sorted into wells containing 1 μl lysis buffer, followed by a heating step and the addition of lysyl-endopeptidase and trypsin to give a total of 2 μl in an enclosed space. Peptides are concentrated in 20 nl nanopackages in an EvoTip device that is similar to a StageTip. Peptides are then subjected to very low flow LC-MS/MS with a novel diaPASEF (parallel accumulation–serial fragmentation) acquisition mode in which peptide ions are released in concentrated packages from the TIMS device into the vacuum system of a quadrupole time-of-flight (TIMS-qTOF) mass spectrometer[26]. These peptide precursors can be fragmented in a highly sensitive manner, either in data-dependent (ddaPASEF) or data-independent (diaPASEF) mode, resulting in very high ion utilization and low rates of missing data[26]. When the latter approach was coupled with very low flow rate chromatography, the authors quantified up to 2083 proteins per HeLa cell using a HeLa DIA spectral library[25].

Capillary electrophoresis coupled with electrospray ionization and high-resolution mass spectrometry (CE-ESI-HRMS) has been demonstrated to have sufficient sensitivity to carry out subcellular proteome analyses[27]. This study analyzed samples that ranged from large, ~250-μm-diameter cells from *Xenopus laevis* embryos to small, ~35-μm-diameter cultured neurons from mouse hippocampus. Analyses on ~18 ng of the extract from the ventral-animal midline or equatorial cells identified 1133 proteins in a 16-cell embryo. Since CE-ESI-HRMS of ~5 ng of protein digest from a left dorsal-animal midline cell identified 722 proteins, this technology has sufficient sensitivity to analyze smaller cells. However, as CE-ESI-HRMS of three single neurons only identified 16, 14, and 7 proteins, respectively, it is likely that manual handling of these cells and their processing in microvials resulted in large adsorptive losses. Together, these studies demonstrate that CE-ESI-HRMS has the potential to provide a valuable complement to nanoLC-based SCP.

As summarized above, reducing sample and digestion volumes to ≤2 μl, limiting carrier spikes in multiplexed SCP technologies to <20-fold, simplifying cell lysis procedures, decreasing HPLC gradients and/or flow rates, optimizing mass spectrometry acquisition parameters, using retention times to bolster peptide sequence identifications, developing software that provides improved processing and analysis of SCP-MS/MS data, and using DIA or DIA-like analyses in place of DDA have all contributed to the substantial advances in SCP analyses over the last few years that now enable up to ~2000 proteins to be quantified in a single HeLa cell extract.

### Integrated approaches/protein "chips"
Even though the studies described above demonstrate remarkable advances in the highly specialized experimental protocols needed for carrying out SCP, there is still a substantial lag in commercially available approaches that can be used for efficient and reproducible lysis, extraction, and trypsin digestion of single-cell proteomes. As SCP methods are constrained by liquid-handling operations, including reagent dispensing, sample aspirating, transferring, and pooling, Liang et al. introduced a fully automated platform termed autoPOTS (automated preparation in one pot for trace samples)[28]. An unmodified, low-cost commercial robotic pipetting platform (i.e., the Opentrons OT-2 liquid handler) was utilized for one-pot sample preparation in low-volume 384-well plates. The digests were analyzed directly from the 384-well plate using a commercial autosampler. Using autoPOTS label-free profiling, the average MS/MS protein identifications increased from 301 for 1 cell to 3347 for 500 HeLa cells[28]. This study demonstrated that the Opentron OT-2 liquid handler could reliably carry out low-μL-scale pipetting with improved sample recovery by minimizing the sample processing volumes in nanowells that reduce nonspecific-binding-related protein/peptide loss.

Woo et al.[29] developed a miniaturized nanoPOTS (N2) chip that improves the isobaric-labeling-based SCP workflow that also reduces the reaction volume to <30 nl and increases capacity to >240 single cells on a single microchip. Based on a cutoff obtained from SCPCompanion, about 50% of the raw MS/MS spectra used in this study provides robust quantification. The analysis of 108 single cells (12 TMT sets) from three murine cell lines (epithelial cells (C10), immune cells (Raw264.7), and endothelial cells (SVEC)) using the N2 chip identified an average of ~7369 unique peptides and ~1716 proteins from each set with at least one valid value in the nine single-cell channels. Cell typing analysis of the three cell populations showed enrichment of each cell type's functions and cell surface markers (membrane proteins)[29].

Recently, Gebreyesus et al. reported a streamlined DIA/MS-based workflow for proteomic analysis down to the single-cell level[30]. They developed a microfluidic device, termed the single-cell integrated proteomics chip (SciProChip), composed of 20 units, which allows samples to be run in parallel. Each unit contains a cell capture, imaging and lysis chamber, a protein reduction, alkylation and digestion vessel, and a peptide desalting column[30] (Fig. 1a). Gebreyesus et al. integrated this microfluidic chip for sample preparation together with DIA-MS and identified on average ~1500 protein groups across 20 single mammalian cells. Further, when analyzing nine cells, the DIA approach provided 2.3-fold higher proteome profiling coverage, lower missing values (i.e., the number of proteins only quantified in one of three triplicate runs was <16%), reproducible quantification, and a 10-fold wider dynamic range than conventional DDA. This method is fully automated and offers streamlined proteomic preparation from cell input to complete sample processing and multiplexing, with high sensitivity and reproducibility for limited input samples, including at the single-cell level[30].

Ctortecka et al. also recently developed the proteoCHIP that, when coupled with the cellenONE robot, provides a universal option for SCP sample preparation at high sensitivity and throughput[31]. This study, which has not yet undergone peer review, describes an automated processing system combining single-cell isolation with picoliter dispensing. The system uses a lysis and trypsin digestion volume of 40 nl that is submerged under a hexadecane layer that prevents evaporation. The polytetrafluoroethylene (PTFE)-based proteoCHIP, which is the size of a standard glass microscope slide, allows direct injection of single cells via a standard autosampler and is designed for simultaneously processing 192 single cells. The low sample and digestion volumes that can be used with the proteoCHIP provide improved reporter ion signal to noise while reducing or eliminating the need for carrier proteomes. Analyses carried out on 218 single HeLa or HEK-293 cells identified an average of 1812 or 1477 proteins/cell in the presence and absence, respectively, of 20x carrier samples[31]. When coupled with the cellenONE® robot, the proteoCHIP provides the first commercial solution for automated, multiplexed SCP.

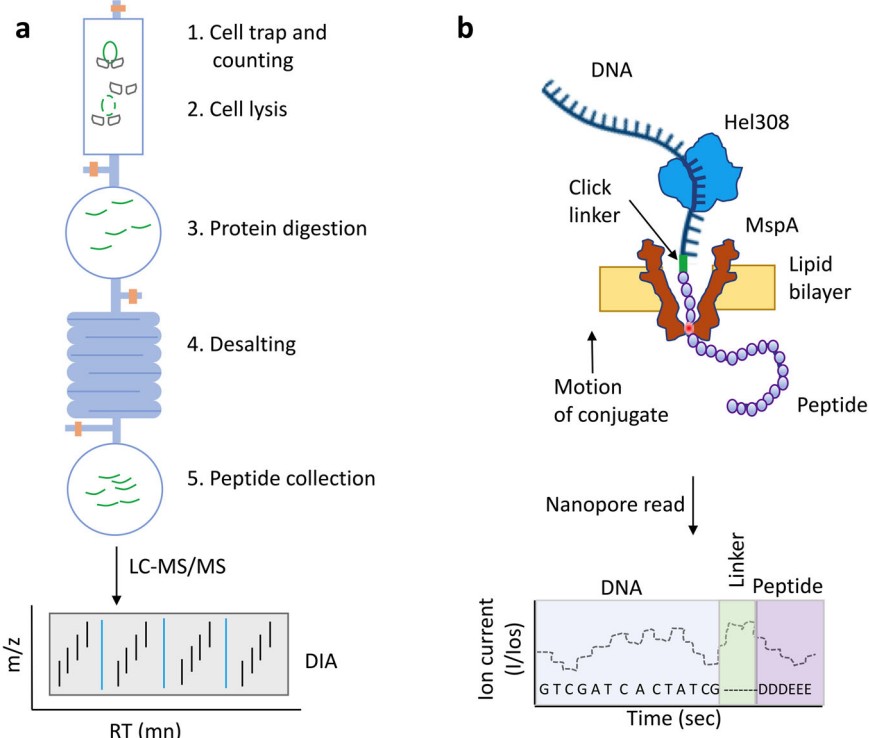

**Fig. 1 Single-cell proteomics methods. a** Schematic of a single unit of a single-cell integrated proteomics chip (SciProChip) and streamlined workflow for nano-proteomics used by Gebreyesus et al. **b** A nanopore device used by Brinkerhoff et al. to read DNA–protein conjugates with single amino acid resolution.

The advances described in this section have made substantial progress toward developing commercially available platforms that are capable of carrying out automated, highly sensitive, and reproducible one-pot lysis, extraction, trypsin digestion, and subsequent LC-MS/MS analyses of single-cell proteomes.

## Single neuron neuropeptides

Neuropeptides are a diverse class of cell-to-cell signaling molecules that are widely expressed throughout the central nervous system, often in a cell-type-specific manner. Recently, a microscopy-guided, high-throughput single-cell matrix-assisted laser desorption/ionization mass spectrometry (MALDI-MS) approach was developed to investigate the neuropeptide heterogeneity of individual neurons in the central nervous system of the neurobiological model *Aplysia californica*[32]. In this study, >26,000 neurons from 18 animals were analyzed, and 866 peptides were assigned to 66 prohormones. This MALDI-MS approach enables the categorization of large cell populations based on single-cell neuropeptide content and is readily adaptable to the study of a range of animals and tissue types.

Another recent study used a CE-TIMS workflow to determine the stereochemistry of pleurin neuropeptides in individual neurons from *Aplysia californica*[33]. The resulting mobility profiles showed that while >98% of the pleurin-derived peptides in individual neuron cell bodies are in the all-L configuration, 44% of the pleurin 2 peptide was in the isomerized, D-configuration. Thus, these findings uncovered an unusual distribution of isomerized neuropeptides in *Aplysia californica* and demonstrate that CE-TIMS MS is capable of interrogating peptide stereochemistry at the single-cell level.

## Outlook

As evidenced by the considerable progress over the last few years, single-cell proteomics is beginning to mature from proof-of concept studies to becoming an important tool in biomedical research that will provide an invaluable complement to single-cell mRNA sequencing. For example, a recent study by Mahdessian et al. compared spatio-temporal relationships in single-cell proteomes and transcriptomes using U2OS cells[34]. The results showed that most cell cycling proteins are regulated post-translationally, rather than by changes at the transcriptomic level. By integrating their spatially resolved proteome map of the cell cycle into the Human Protein Atlas, a valuable resource was created that will support human cell cycle and cell proliferation studies.

The various methods described above, while becoming more technically complex, generally represent the optimization of MS-based workflows, instrumentation, and data analysis. Despite the impressive advances over the last few years in MS-based SCP, technologies for characterizing proteins still lag far behind those for nucleic acids, which are characterized by extremely high sensitivity, dynamic range, and throughput[35]. The ability to characterize proteins and, especially, the manifold larger numbers of proteoforms at nucleic acid levels would help address critical biomedical and biological challenges, such as finding the biomarkers needed to provide improved diagnosis, classification, and prognoses of diseases, increase the depth and speed of proteome quantification, and provide new approaches to SCP. In response to this need, significant efforts have been made to improve protein sequencing technologies by adapting technologies used for high-throughput nucleic acid sequencing, with a particular focus on developing practical methods for single-molecule protein sequencing (SMPS)[35], see Box 1. As described below, such

---

**Box 1 ▌ Future developments**

SMPS technologies can be divided into three categories: sequencing by degradation (e.g., mass spectrometry or fluorosequencing), sequencing by transit (e.g., nanopores or quantum tunneling), and sequencing by affinity (e.g., DNA hybridization-based approaches)[35, 36]. In the fluorosequencing approach described by Swaminathan et al., cysteine and lysine residues are labeled in peptide samples prior to immobilization of the labeled peptides onto a glass surface and imaging by total internal reflection microscopy (TIRF) to monitor reductions in each molecule's fluorescence following consecutive rounds of Edman degradation[37]. The limited fluorescent sequencing data on each molecule was then used to assign each peptide to its parent protein in a reference database. Swaminathan et al. demonstrated the method on synthetic and naturally derived peptide molecules at zeptomole-scale levels. In addition, they also fluorescently labeled phosphoserines in peptides and demonstrated single-molecule, positional readout of the phosphorylated sites[37]. Further improvements in the technology should enable analyses of increasingly complex proteomic mixtures.

As a completely different alternative approach to sequencing peptides that also may ultimately be applicable to SCP, Brinkerhoff et al. described a nanopore-based approach called single-molecule peptide fingerprinting[38] that was inspired by single-molecule DNA sequencing technology. When a single strand of DNA is slowly passed through a protein nanopore embedded in a thin membrane, each base partially blocks an electrical current that is being carried by ions through the nanopore. This generates a series of ion current steps corresponding to bases residing in the pore, ultimately providing the DNA base sequence[36, 38]. Brinkerhoff et al. developed a similar system in which an 80-nucleotide DNA–26-amino acid peptide conjugate is pulled through the biological nanopore in MspA by the Hel308 DNA helicase (Fig. 1b). As the DNA helicase "walks" along the DNA, the ratcheting motion of the peptide portion of the conjugate in the nanopore generates a characteristic steplike pattern in the ion current signal, with differential ion currents being monitored to identify differences in the amino acid sequence along the peptide backbone. This approach is very sensitive and is able to discriminate single amino acid substitutions in single reads. Further, the method allows for rewinding and rereading the same peptide multiple times, which reduces the misreading error rate to <1 in $10^6$ when using more than ~30 rereads of an individual peptide containing a single amino acid variant[38].

---

techniques and others yet to be developed hold great promise for SCP to take a more central stage in both basic and applied research applications in the future.

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

## Acknowledgements

We especially thank the reviewers for carefully reading our manuscript and for making excellent and highly detailed suggestions that greatly improved this publication. We acknowledge support from the NIH (Yale/NIDA Neuroproteomics Center grant DA018343). Research in the laboratory of ACN is supported by NIH (AG047270, AG062306, AG066508, DA018343) and the State of Connecticut Department of Mental Health and Addiction Services.

## Author contributions

Conceptualization, M.S.M., K.R.W., and A.C.N.; methodology, M.S.M., K.R.W., and A.C.N.; literature investigation M.S.M., K.R.W, and A.C.N.; writing—original draft preparation, M.S.M., K.R.W. and A.C.N.; writing—review and editing, M.S.M., K.R.W., and A.C.N.; funding acquisition, K.R.W. and A.C.N.

## Competing interests

The authors declare no competing interests.
