## [Peer Review File · Communications Biology]

Uncovering biology by single-cell proteomicsReviewers' comments:

Reviewer #1 (Remarks to the Author):

In this manuscript, the authors are discussed several novel approaches about single-cell proteomics. The manuscript might be accepted after major revisions.

One of the missing aspects, which are relevant in the field, it is related about bioinformatics tools and software design and development. I am strongly recommend to the authors, also to include these points in the revised version.

Another recent advance in the field of single proteomics, it is also missing, and has to be included and discussed because it is describing spationtemporal dimension in single cell proteomics (Mahdessin et al. Nature 2021, 590, 649-654).

Reviewer #2 (Remarks to the Author):

Mansuri et al survey recent advances in single cell proteomics with an emphasis on mass spectrometry centered approaches. Overall, the review is excellent and well written and they did hit on the more visible studies and on the relevant topics such as sample handling. They did miss a few studies of relevance. Nemes and Dovichi have separately analyzed developing xenopus embryos at the single cell level. Nemes has also done SCP on neurons and, of course, Jon Sweedler has been doing single cell analyses for 30 years although not so much on the proteome level, but certainly in terms of neurobiology he has been leader in the analysis of neuropeptides at the single neuron level. As you mention nanopore sequencing do you also want to mention Ed Marcotte's single molecule Edman sequencing? There are a couple of papers, but the technology has been spun out into a company. It's likely to be an interesting alternative for SCP.

In summary, nicely done with some notes about a few misses.

Reviewer #3 (Remarks to the Author):

It should first be noted that the review is not authoritative, as nobody in the author list appears to have contributed to the field they are reviewing. This is not necessarily disqualifying, but it is clear that the authors' understanding of the field and how it progressed is lacking. At a minimum, several issues with the manuscript should be addressed prior to acceptance:

1. In the 2nd paragraph, the authors report on a 'typical' detection limit for mass spectrometers of 290 attomoles. Obviously, single cell proteomics require highly sensitive instrumentation, and it is more informative to report achievable detection limits, not typical detection limits. Indeed, if the lowest detection limit we could achieve were 290 amol, this would not be encouraging for SCP. There are numerous examples of zeptomole detection limits that should be reported, including: DOI: 10.1021/ac030096q; 10.1002/anie.201308139; 10.1039/C0AN00253D; 10.1074/mcp.M114.045724; 10.1039/C0AN00253D.
2. In the 3rd paragraph, the authors provide as a rationale for SCoPE-MS that the carrier reduces losses on-column. This may occur, but the authors of the work did not demonstrate this experimentally. The main rationale for SCoPE-MS is that a bulk sample labeled with one TMT reagent can trigger MS2 and enable quantification of single cells in the TMT set based solely on their reporter ion signal. The authors neglect this completely.
3. nPOP is an adaptation of nanoPOTS and was reported years after nanoPOTS. Indeed, the various groups working in the field learn from each other and adapt the methods of one another, which moves the field forward. By presenting all of the Slavov group's developments first, this provides a revisionist view of how the field developed.
4. SCoPE MS does not report >1000 proteins per cell but rather 1526 proteins across all measured

cells (Figure 3). These are two different things, and the authors should correct this.

5. Along these same lines, nanoPOTS was not developed to “circumvent the issues of SCoPE-MS”, but was rather developed for label-free proteomics concurrently with SCoPE-MS. Please note the submission and publication dates of both to convince yourself of this: NanoPOTS was submitted for publication in April 2017 and published in February 2018. NanoPOTS as applied to single cells (10.1002/anie.201802843) was submitted in March 2018 and published in June 2018. The first preprint of SCoPE MS appeared in 2017 and the work was submitted for publication in February 2018 to Genome Biology and published online in October 2018. The inaccurate reporting does a great disservice to those who brought about the SCP field.

6. The major issue with SCoPE MS is not that it is technically challenging, but rather that general issues with TMT such as precursor coisolation/ratio compression are especially severe for SCP. Isotopic contamination is also a significant limitation. In addition, the ‘carrier proteome effect’ as reported by Chris Rose further reduces quantitative accuracy when a carrier channel is used. The authors neglect all of this.

7. The ProteoChip as developed by Cellenion and the Mechtler group receives no word in this report. This should be remedied, as this is the first commercial approach for SCP and would fit nicely alongside other efforts to make SCP more accessible.

Reviewers' comments:

We thank the reviewers for their constructive criticisms. Our responses and edits to each point are shown below in bold font.

Reviewer #1 (Remarks to the Author):

In this manuscript, the authors discussed several novel approaches about single-cell proteomics.

The manuscript might be accepted after major revisions.

1. One of the missing aspects, which are relevant in the field, it is related about bioinformatics tools and software design and development. I am strongly recommend to the authors, also to include these points in the revised version.

Response: We appreciate your suggestion. We have modified the manuscript and added discussion of bioinformatics tools and software designed to support single cell proteomics studies, including DO-MS, DART-ID, IceR, SCPCompanion, SCeptre, and PASEF.

2. Another recent advance in the field of single proteomics, it is also missing, and has to be included and discussed because it is describing spationtemporal dimension in single cell proteomics (Mahdessin et al. Nature 2021, 590, 649-654).

Response: Thank you very much for your suggestion. We have acknowledged the importance of other single cell proteomics studies and noted this study in the revised manuscript.

Reviewer #2 (Remarks to the Author):

Mansuri et al survey recent advances in single cell proteomics with an emphasis on mass spectrometry centered approaches. Overall, the review is excellent and well written and they did hit on the more visible studies and on the relevant topics such as sample handling.

1.They did miss a few studies of relevance. Nemes and Dovichi have separately analyzed developing xenopus embryos at the single cell level. Nemes has also done SCP on neurons and, of course, Jon Sweedler has been doing single cell analyses for 30 years although not so much on the proteome level, but certainly in terms of neurobiology he has been leader in the analysis of neuropeptides at the single neuron level.

Response: Thank you very much for your suggestions. We have acknowledged the importance of other single cell proteomics studies and added discussion of your suggested studies in the revised manuscript.

2. As you mention nanopore sequencing do you also want to mention Ed Marcotte's single molecule Edman sequencing? There are a couple of papers, but the technology has been spun out into a company. It's likely to be an interesting alternative for SCP.
In summary, nicely done with some notes about a few misses.

Response: Thanks for your suggestion. We have mentioned nanopore sequencing method in the revised manuscript and noted the above mentioned technology.

Reviewer #3 (Remarks to the Author):

It should first be noted that the review is not authoritative, as nobody in the author list appears to have contributed to the field they are reviewing. This is not necessarily disqualifying, but it is clear that the authors' understanding of the field and how it progressed is lacking. At a minimum, several issues with the manuscript should be addressed prior to acceptance:

Response: We thank the reviewer for their constructive criticisms.

1. In the 2nd paragraph, the authors report on a 'typical' detection limit for mass spectrometers of 290 attomoles. Obviously, single cell proteomics require highly sensitive instrumentation, and it is more informative to report achievable detection limits, not typical detection limits. Indeed, if the lowest detection limit we could achieve were 290 amol, this would not be encouraging for SCP. There are numerous examples of zeptomole detection limits that should be reported, including: DOI: 10.1021/ac030096q; 10.1002/anie.201308139; 10.1039/C0AN00253D; 10.1074/mcp.M114.045724; 10.1039/C0AN00253D.

Response: We have revised the text and added the achievable detection limits cited in the Shen et al (2004) PMID: 14697044 reference that was provided by the reviewer. Of the five references provided by the reviewer, we believe the Shen et al reference is the most relevant as it used LC/MS (as opposed to CZE/MS in the Sun et al (2013) PMID: PMC3904452 reference) and it also was carried out on a whole proteome digest (as opposed to the Li et al. (2015) PMID: PMC4458728 study that was carried out on rare cells in whole blood).

2. In the 3rd paragraph, the authors provide as a rationale for SCoPE-MS that the carrier reduces losses on-column. This may occur, but the authors of the work did not demonstrate this experimentally. The main rationale for SCoPE-MS is that a bulk sample labeled with one TMT reagent can trigger MS2 and enable quantification of single cells in the TMT set based solely on their reporter ion signal. The authors neglect this completely.

Response: We have revised the text as suggested.

3. nPOP is an adaptation of nanoPOTS and was reported years after nanoPOTS. Indeed, the various groups working in the field learn from each other and adapt the methods of one another, which moves the field forward. By presenting all of the Slavov group's developments first, this provides a revisionist view of how the field developed.

Response: In our revised manuscript we have largely put the SCP studies in chronological order based on their publication dates.

4. SCoPE MS does not report >1000 proteins per cell but rather 1526 proteins across all measured cells (Figure 3). These are two different things, and the authors should correct this.

Response: This error has been corrected.

5. Along these same lines, nanoPOTS was not developed to "circumvent the issues of SCoPE-MS", but was rather developed for label-free proteomics concurrently with SCoPE-MS. Please note the submission and publication dates of both to convince yourself of this: NanoPOTS was submitted for publication in April 2017 and published in February 2018. NanoPOTS as applied to single cells

(10.1002/anie.201802843) was submitted in March 2018 and published in June 2018. The first preprint of SCoPE MS appeared in 2017 and the work was submitted for publication in February 2018 to Genome Biology and published online in October 2018. The inaccurate reporting does a great disservice to those who brought about the SCP field.

Response: We have put the SCP studies in chronological order based on their publication dates and believe that any errors have been corrected.

6. The major issue with SCoPE MS is not that it is technically challenging, but rather that general issues with TMT such as precursor coisolation/ratio compression are especially severe for SCP. Isotopic contamination is also a significant limitation. In addition, the 'carrier proteome effect' as reported by Chris Rose further reduces quantitative accuracy when a carrier channel is used. The authors neglect all of this.

Response: All of the above issues with SCoPE MS have been included in the revised manuscript

7. The ProteoChip as developed by Cellenion and the Mechtler group receives no word in this report. This should be remedied, as this is the first commercial approach for SCP and would fit nicely alongside other efforts to make SCP more accessible.

Response: We have added this important development and noted that it is the first commercial approach for SCP.

REVIEWERS' COMMENTS:

Reviewer #3 (Remarks to the Author):

The reviewers have addressed major concerns and the manuscript is acceptable for publication.